# Impact of Fermentation and Phytase Treatment of Pea-Oat Protein Blend on Physicochemical, Sensory, and Nutritional Properties of Extruded Meat Analogs

**DOI:** 10.3390/foods9081059

**Published:** 2020-08-05

**Authors:** Aleksei Kaleda, Karel Talvistu, Martti Tamm, Maret Viirma, Julia Rosend, Kristel Tanilas, Marie Kriisa, Natalja Part, Mari-Liis Tammik

**Affiliations:** Center of Food and Fermentation Technologies, Akadeemia tee 15a, 12618 Tallinn, Estonia; karel.talvistu@tftak.eu (K.T.); martti@tftak.eu (M.T.); maret@tftak.eu (M.V.); julia@tftak.eu (J.R.); kristel@tftak.eu (K.T.); marie@tftak.eu (M.K.); natalja@tftak.eu (N.P.); mariliis.tammik@tftak.eu (M.-L.T.)

**Keywords:** meat analog, extrusion, fermentation, phytic acid

## Abstract

Plant materials that are used for the production of extruded meat analogs are often nutritionally incomplete and also contain antinutrients, thus there is a need to explore alternative plant proteins and pre-treatments. This study demonstrates application of phytase and fermentation to a pea-oat protein blend with a good essential amino acid profile and subsequent texturization using extrusion cooking. Enzymatic treatment reduced the content of antinutrient phytic acid by 32%. Extrusion also degraded phytic acid by up to 18%, but the effect depended on the material. Differences in physicochemical, sensorial, and textural properties between untreated and phytase-treated extruded meat analogs were small. In contrast, fermented material was more difficult to texturize due to degradation of macromolecules; physicochemical and textural properties of extrudates were markedly different; sensory analysis showed enhancement of flavor, but also detected an increase in some unwanted taste attributes (bitterness, cereal and off-taste). Phytic acid was not degraded by fermentation. Analysis of volatile compounds showed extrusion eliminated volatiles from the raw material but introduced Maillard reaction products. Overall, phytase treatment and fermentation demonstrated the potential for application in extruded meat analogs but also highlighted the necessity of optimization of process conditions.

## 1. Introduction

Demand for plant protein sources to replace meat is growing due to the concerns of environmental sustainability, human health, and ethical reasons [1]. Traditional plant-based meat substitutes such as tofu or seitan have limited acceptance in Western diets, however, meat analogs in the form of textured vegetable proteins are rapidly gaining popularity [2]. Extrusion cooking is the most widely used technology for protein texturization [3]. Commonly used soybean protein and wheat gluten form good meat-like texture, but their limitation is significant allergenicity and in the case of gluten, the poor nutritional profile of essential amino acids [4]. This makes the producers consider alternatives—pea, oat, fava bean, and other proteins that are currently found in commercial products [5]. Oat and pea proteins are similarly limited in some essential amino acids, however, their mixture has an improved nutritional profile due to their complementarity [6].

A major problem concerning plant materials is that they may contain high amounts of antinutritional components including phytic acid, lectins, saponins, tannins, and enzyme inhibitors [7]. Antinutrients interfere with mineral bioavailability and digestibility of proteins and carbohydrates; can cause gastroenteritis and diarrhea [8]. Frequently a single plant contains multiple antinutrient compounds.

Phytic acid (*myo*-inositol hexakisphosphate, IP6) and its salts (phytates) are the principal storage form of phosphate and inositol in plants. In legumes, cereals, oilseeds, and nuts it is typically present at levels of 0.5–5%. Phytic acid is regarded as an antinutrient due to its strong binding affinity to Ca, Fe, Zn, Na, K, Cu, Co, and Mg ions. High concentrations of phytic acid can limit the bioavailability of important mineral micronutrients, especially in populations predominantly reliant on legume and cereal diets [9].

Lectins, also known as hemagglutinins, are glycoproteins that are present in legumes and grains. Lectins can bind to mucosal cells and interfere with digestion and nutrient absorption [8].

Lectins are almost completely degraded by heat treatment conditions that are applied during extrusion [10], however, phytic acid is more resistant. Reports on the degree of degradation of phytic acid by extrusion range from no effect to almost complete elimination [11,12]. This discrepancy potentially arises from different tested materials and extrusion parameters such as temperature and moisture. It should also be noted that some of the phytic acid quantification methods are not able to distinguish between different forms of inositol with fewer phosphate groups and therefore would not detect changes in the composition of inositol phosphates [13]. This is important because it was shown that extrusion can partially degrade IP6 to IP5 that has weaker antinutritional properties [14,15].

Alternative treatments have to be employed to reduce phytic acid content. Enzymatical treatment of plant materials or direct supplementation of phytase have found use in the feed industry. Multiple studies demonstrate an improvement of feed nutritional value and growth performance in poultry and pigs [16,17].

Fermentation has also been studied for phytic acid degradation and many phytase-producing bacteria, yeasts, and fungi have been identified [18,19]. Fermentation was shown to reduce phytic acid to a different degree depending on the substrate and microorganisms. Fermentation was applied to a variety of grains, bread making, beer brewing, traditional foods, and soy milk [18]. Yet, there are no studies of fermentation for the production of extruded meat analogs.

Therefore, the objective of this study was to apply phytase and fermentation treatments to a mixture of pea and oat proteins to examine nutritional, physical, and sensorial changes of extruded meat analogs.

## 2. Materials and Methods

### 2.1. Materials

Commercial oat protein concentrate PrOatein^®^ was obtained from Lantmännen Oats (Kimstad, Sweden). The proximate composition was 54.4% protein, 24.7% carbohydrates, and 18.2% fat on a dry weight basis (dwb).

Commercial pea protein isolate was obtained from Caremoli S.p.A (Monza, Italy). Powder protein content was 84.7%, and 9.3% fat dwb.

A 70:30 blend of pea and oat protein powders was used throughout the experiments, where 75.6% was protein, 7.4% carbohydrates, and 12.0% fat. Correspondingly, 59.3% of the mixture were proteins from pea and 16.3% from oat, a 3.6:1 ratio.

The phytase, Phyzyme^®^ XP 5000L, was kindly provided by Danisco Animal Nutrition (Dupont, Wiltshire, UK).

### 2.2. Bacterial Cultures and Preparation of Inoculum

The commercial lyophilized starter cultures from DuPont Danisco^®^ VEGE 022 (contains *Streptococcus thermophilus*, *Lactobacillus delbrueckii* subsp. bulgaricus, *Lactobacillus acidophilus*, *Bifidobacterium lactis*, *Lactobacillus plantarum*), VEGE 053 (*Streptococcus thermophilus*, *Lactobacillus delbrueckii* subsp. bulgaricus, *Lactobacillus acidophilus*, *Bifidobacterium lactis*, *Lactococcus lactis*), VEGE 061 (*Streptococcus thermophilus*, *Lactobacillus delbrueckii* subsp. bulgaricus, *Lactobacillus acidophilus*, *Bifidobacterium lactis*, *Lactobacillus paracasei*), and *Weissella* sp. from the in-lab collection were used in the experiment. Additionally, *Lactobacillus plantarum* DPPMAB24W was kindly provided by the Department of Plant Protection and Applied Microbiology of the University of Bari (Bari, Italy).

The *Lactobacillus plantarum* DPPMAB24W and *Weissella* sp. were cultured in MRS broth at 30 °C until the late exponential phase (ca. 20 h), washed twice in 1x PBS and maintained as frozen stock in 50% glycerol at −80 °C. The inocula were prepared from the stocks and commercial lyophilized starter cultures by diluting in 0.85% NaCl solution. The inoculum size was 1% at 10^7^ CFU/mL initial cell density in the sample.

### 2.3. Fermentation

Fermentation experiments were carried out with pea-oat powder suspensions in tap water at 15% dry matter with an additional 1 g L^−1^ of sucrose to assist fermentation.

Small-scale fermentation (100 mL) was conducted at 30 °C or 40 °C for 22 h in the iCinac system (AMS S.r.l, Rome, Italy) to measure the acidification activity of lactic acid bacteria. In parallel, the growth of bacteria in sealed 2 mL vials was also monitored by using a 48-channel isothermal batch microcalorimeter TAM IV Thermal Activity Monitor (TA Instruments, New Castle, DE, USA). The output of the microcalorimeter is a heat flow curve describing the evolution of the fermentation process [20].

Large-scale batch bioprocess was conducted in 7 L “BioBench” fermenter (Applikon Biotechnology, Delft, The Netherlands) under aerobic conditions. The fermentation process was controlled by the cultivation control software “BioXpert” and ADI 1010 bio-controller (Applikon). Powder suspension (5 L) was fermented by VEGE 053 at 30 °C for 24 h under slow continuous stirring (150 rpm), and the pH was monitored. Afterward, the fermented material was frozen at −40 °C and lyophilized.

### 2.4. Phytase Treatment

The pea-oat powder was suspended in tap water to 15% dry matter and the experiment was carried out without an additional pH adjustment (pH = 6.7). The suspension was heated to 40 °C under continuous stirring conditions in a 10 L Limitech lab mixer (Limitech A/S, Aabybro, Denmark). Thereafter, 1.5% of phytase (Phyzyme^®^ XP) was added to the suspension based on the dry matter content. The phytase was added in excess to ensure phytic acid degradation during 4-h incubation at 40 °C under minimum agitator speed. After 4-h incubation the suspension was collected, frozen at −40 °C and then lyophilized. The enzyme retained in lyophilized powder was inactivated by heat in further extrusion experiments or during phytic acid determination.

### 2.5. Phytic Acid Analysis

Phytic acid content was assessed by two methods.

Total phytic acid content was measured using a phytic acid assay kit from Megazyme according to the manufacturer’s instructions. This kit quantifies phosphorus released from *myo*-inositol phosphates by phytase and alkaline phosphatase. This method assumes that the content of other phosphates is insignificant.

Phytic acid and other inositol phosphates (*myo*-inositol mono-, bis-, tris-, tetrakis-, and pentakisphosphates) were separated and quantified in various matrices using LC-MS-TOF combined with stable isotope dilution assay. The LC-MS-TOF analyses were performed as described by Rougemont et al. (2016) [21] with some modifications. IP1–IP6 were extracted from food matrices after 3 h extraction with cold 0.5N HCl at room temperature [22]. U-13C-labeled maize extract (IsoLife, Wageningen, Netherlands) was used as a source of labeled phytates.

### 2.6. Rheology

The viscosity of fermented samples was measured at 25 °C by a shear rate sweep (0.01–100 s^−1^) using Physica MCR301 rheometer (Anton Paar GmbH, Graz, Austria) equipped with a Peltier temperature control unit C-PTD200 and a coaxial cylinder measuring system CC27 (outer and inner diameters 28.92 and 26.66 mm, respectively).

### 2.7. Extrusion

Control, phytase-treated, and fermented pea-oat powders were sequentially processed in a twin-screw extruder KETSE 20/40 (Brabender GmbH, Duisburg, Germany). The screw was configured to impart high shear (Figure 1). Extrusion conditions were determined in preliminary tests. Powder and water mass flow rates were set constant by calibrating feeder and pump with each powder or tap water, respectively. The screw speed was set to 600 rpm, dough moisture to approximately 30% (1200 rpm and 25% in the case of fermented powder), total feed rate 5 kg h^−1^, and temperatures of the six heating zones were set to 40/70/130/150/140/140 °C, respectively. Die opening diameter was 2 mm. The extrudates were collected after process stabilization as evidenced by constant product temperature, die pressure, and screw torque recorded by the extruder every second. Collected samples were allowed to cool down, then dried for 2 h at 40 °C until water activity reached below 0.6 and packed into zip-lock bags until further analysis.

The specific mechanical energy (SME) was calculated as described by do Carmo et al. (2019) [23]. The mass flow rate was calculated by collecting the extrudate for three minutes and then recording the mass of used water, the mass of dried extrudate, the moisture content of powder, and the moisture content of dried extrudate. A blank extrusion experiment with water was performed at different screw speeds to record the base torque.

### 2.8. Texture Profile Analysis

Texture profile analysis was performed using TA.XT2i Texture Analyzer (Stable Micro-Systems, Godalming, UK) equipped with a 75 mm flat probe and 5 kg load cell. Dry samples were rehydrated in tap water at 60 °C for 7 min and then excess water was removed with paper towels. Extruded pieces (0.9 g) were compressed twice in transversal to the extrusion flow direction by 75% at 1 mm s^−1^ probe speed and 2 s holding time between compressions. Hardness, cohesiveness, springiness, resilience, and chewiness were calculated by the texture analyzer software.

### 2.9. Oil Holding Capacity, Water Holding Capacity, and Water Solubility Index

The methods for measuring oil holding capacity (OHC), water holding capacity (WHC), and water solubility index (WSI) were adapted from Stojceska et al. (2009) [24]. Briefly, dry extrudates were ground into a fine powder in a small coffee grinder and 1 g of powder was resuspended in 10 mL of distilled water or rapeseed oil and gently mixed with a tube rotator for 30 min at room temperature. Thereafter, samples were centrifuged at 3000× *g* for 15 min at 25 °C, supernatant carefully decanted and the remaining gel weighed. WHC and OHC were expressed as the weight of water or oil in grams held by 1 g of powder. Water supernatant was collected and lyophilized to calculate WSI, expressed as the percentage of dry solids in the supernatant to the initial powder weight.

### 2.10. Color

The color of powders and ground extrudates was measured using Chroma Meter CR-400 (Konica Minolta Inc., Tokyo, Japan). The instrument was calibrated with a white tile and color was expressed in CIELAB color space.

### 2.11. Amino Acid Profile

Total and free amino acid contents were determined according to the method of Lahtvee et al. (2014) [25].

### 2.12. Sensory Profile

A trained panel of eight assessors evaluated the extrudates. The panelists had at least two years of experience with sensory analysis. An additional training session was carried out with all the samples prior testing session to familiarize with the products and specify attributes for the assessment. Samples were rehydrated with tap water at 60 °C for 10 min, blotted dry with a paper tissue, and divided as separate portions into 40 mL transparent plastic cups. The samples were evaluated in two parallels. All the samples were coded with a three-digit number, and the order of samples was randomized according to Williams’ Latin square design. The assessment was performed in a sensory analysis room in accordance with ISO 8589:2007. Sensory attributes were graded on a 0–9 scale, where 0 means none, 1—very weak, 5—moderate, and 9—very strong.

### 2.13. Volatile Compounds

Volatile compounds were analyzed following the Taivosalo et al. (2019) [26] headspace solid-phase microextraction gas chromatography mass spectrometry method with the following adaptations. Vials containing 1 g of a sample were preincubated at 50 °C for 5 min. Volatile compounds were extracted from the headspace with a fiber for 20 min under stirring at 50 °C and eluted with a temperature ramp from 40 °C to 280 °C at a rate of 7.5 °C min^−1^ with an additional holding time of 3 min in the end. Internal standard 4-methyl-2-pentanol at 200 ppb was used and amounts were expressed as ppb in internal standard equivalent. The results were adjusted according to the sample dry matter.

### 2.14. Statistical Analysis

Small-scale fermentation in iCinac and microcalorimeter, amino acids, inositol phosphates, volatile compounds, oil holding capacity, water holding capacity, water solubility index, and color were measured in triplicate. Texture profile analysis was done in 12 replicates. Sensory assessment and rheology were performed in duplicate.

Statistical analysis and visualization were done in R version 4.0.0 (R Foundation for Statistical Computing, Vienna, Austria). Package ‘agricolae’ 1.3-2 was used for Tukey’s honestly significant difference test at alpha 0.05. Partial least squares discriminant analysis was performed with package ‘mixOmics’ 6.11.33. Package ‘sparsepca’ 0.1.2 was used for calculating sparse principal components.

## 3. Results and Discussion

### 3.1. Small Batch Fermentation

Starter cultures for fermentation of the pea-oat protein blend were selected according to their ability to ferment plant material and possibly degrade phytates. Chosen commercial starters Danisco VEGE 022, 053, and 061 contain lactic acid bacteria *Lactobacillus plantarum*, *Lactobacillus delbrueckii*, and *Lactobacillus acidophilus*, some strains of which were previously shown to have phytase activity [18]. Additionally, pure cultures of *Weissella* sp. and *Lactobacillus plantarum* DPPMAB24W were selected for their potential to degrade antinutrients [18,27].

All chosen starters were able to grow in the pea-oat protein suspension supplemented with sucrose, as evidenced by the microcalorimetry heat flow curves (Figure 2A) and pH drop (Figure 2B). In all the cases pH has decreased approximately by one unit from 6.7 to 5.7. Spontaneous sample and VEGE starters had multiple heat flow peaks associated with the active growth phases of different bacteria of the starter consortium. Fermentation at 40 °C concluded in 15 h, but at 30 °C, it finished in 22 h, except for VEGE 022, which at the end of the experiment was accelerating its heat production indicating the beginning of an active growth phase.

Evaluation of both heat flow and pH curves provides a deeper understanding of the fermentation process. By looking only at the pH curve (Figure 2B), one might conclude that *L. plantarum* culture did not grow in the medium, as it is not different from the spontaneous curve. But evaluation of the heat flow curves (Figure 2A) confirms that *L. plantarum* did grow and even had the shortest lag-phase of all the bacteria growing at 30 °C.

The total phytic acid content of the fermented samples measured by the Megazyme kit is presented in Table 1. According to this method, phytic acid content unexpectedly increased after fermentation, particularly in the case of *L. plantarum* and spontaneous sample. These peculiar results could be attributed to the non-specificity of the method, but further investigation is needed to explain this inconsistency.

A more targeted LC-MS method was applied to verify the results of the Megazyme kit. Table 2 shows the content of IP6, IP5, and IP1 that were detected; IP4 was below the quantification limit. In contrast to the Megazyme kit, this method reveals that there is no difference in inositol phosphates content between fermented and non-fermented samples. Multiple studies have demonstrated degradation of phytates by fermentation with lactic acid bacteria [18], but certain phytase-producing strains are necessary. In this experiment, none of the starters selected for testing had phytase activity.

### 3.2. Large Batch Fermentation

The recommended incubation temperature for VEGE starters is 40 °C, however, at 30 °C the viscosity of the fermented suspension was visibly higher due to the production of exopolysaccharides (EPS). Lactic acid bacteria can utilize sucrose for the production of a variety of EPS, which can improve the texture of final products [28]. Viscosity of control, VEGE 022, 053, and 061 at 0.015 s^−1^ strain was 0.44 ± 0.02, 0.44 ± 0.01, 0.74 ± 0.08, and 0.52 ± 0.03 Pa s, respectively. Thus, the starter for fermentation in a bigger batch, VEGE 053, was chosen according to its ability to produce EPS as none of the starters degraded phytates.

Bigger batch fermentation with VEGE 053 was carried out in a 7 L bioreactor. The pH curve, in this case, was similar in shape to the small fermentation, but the decrease occurred faster. In small batch pH dropped from 6.7 to 5.7 in approximately 19 h, but in bigger batch in 12 h. This difference can be attributed to the mixing applied in the bigger batch which improved mass transfer and consequently fermentation speed, whereas in small batches powder precipitated and created a thick layer.

### 3.3. Phytase Treatment

One possibility to reduce phytic acid content is by applying exogenous microbial phytases. Depending on the enzyme source the optimal hydrolysis conditions can vary a lot [29]. A phytase used in this study, Phyzyme XP, was sourced from an *Escherichia coli* and expressed in *Saccharomyces pombe*. The highest activity of this enzyme is at low pH values (pH = 4.5) and at 55 °C [29]. However, for this experiment the incubation temperature (40 °C) was chosen based on the temperature applied also in fermentation experiments. Additionally, the aim was to avoid the effect of high temperature on proteins during incubation. According to Figure 2A spontaneous fermentation began after 4 h, thus limiting phytase treatment time. To avoid the effect of added acid on the extruded product pH was not adjusted, and as the reaction conditions were not optimal, then the phytase was applied in excess. Used incubation conditions allow a better comparison between control and fermented samples.

Different studies demonstrate that various exogenous phytases can efficiently reduce phytic acid content from 20% to 100%, depending on reaction conditions [30,31]. The results in this study show that under used reaction conditions up to 30% of phytic acid could be degraded. The total inositol phosphates content (IP6, IP5, and IP1) in non-treated and enzymatically treated samples was 1.47% and 1.03% dwb, respectively. However, optimized incubation conditions would allow a higher degree of phytic acid degradation.

### 3.4. Extrusion of Meat Analogs

Fermented and enzymatically treated suspensions were lyophilized as accurate dosing into the extruder in hydrated form was not possible for technical reasons. Control and treated powders were extruded and collected in sequence when the extruder reached a steady state. Fibrous meat-like textures were obtained from all powders, as seen in Figure 3.

Recorded extrusion conditions are presented in Table 3. The phytase-treated powder was extruding similarly to the control, in contrast, the fermented powder did not initially form fibrous texture at the same extrusion settings, thus treatment intensity was increased by slightly lowering the dough moisture and doubling the screw speed from 600 rpm to 1200 rpm. Fermented powder created significantly lower pressure at the die even at a reduced moisture content as a result of bacterial enzymatic activity that degraded proteins and carbohydrates, which in turn lowered the dough viscosity [32]. Consequently, higher screw speed was necessary in the case of fermented powder to achieve SME similar to the control powder.

In comparison to expanded snacks produced by do Carmo et al. (2019) [23] from pea and oat fractions with an analogous extruder setup, meat analogs (Table 3) extruded at an order of magnitude lower pressure, but achieved higher SME. The values for expanded snacks were in the range 98–213 bar and 88–145 Wh kg^−1^. Such divergence in pressure can be attributed to the low viscosity of the dough mainly caused by higher moisture but also by different powder composition, i.e., higher protein content. Previous studies have demonstrated that the addition of proteins in the form of isolates typically reduced SME, however, in this work the screw had multiple kneading and reverse elements that imparted higher mechanical treatment [33].

### 3.5. Physicochemical Properties

The physicochemical properties of powders and corresponding extrudates are listed in Table 4. The WHC, WSI, and OHC characterize interaction with water or oil and are important when extruded materials are processed into final products. Protein constituted 75.6% of the pea-oat blend, thus changes in physicochemical properties can be attributed mostly to it.

Water binding depends on the availability of polar hydrophilic groups and is strongly influenced by changes in the composition and conformation of proteins. Phytase treatment of the control powder did not change WHC as the enzyme was not acting upon proteins and the process was at mild conditions leaving proteins intact. Relatively short time also prevented molecular changes caused by microbiological action. This contrasts with the fermented powder that has 33% lower WHC. In addition to bacterial proteolysis, other factors could have contributed to the decrease of WHC. Protein charge and conformation are influenced by pH, which fell by one unit during fermentation. Degradation and consumption of carbohydrates by bacteria could also have contributed to the WHC, as these molecules bind a lot of water. High temperature and shearing during extrusion denature proteins making hydrophilic groups less available and that explains lower WHC values of extrudates. However, these values were not significantly different across all three samples and also were similar to the WHC of the fermented powder. One explanation is that all these treatments were severe enough to be bound by the lowest possible WHC value [34].

Oil holding capacity is determined by the nonpolar side chains of proteins but is also dependent on the physical entrapment of oil and can be explained by the material microstructure [34]. OHC of the control powder and control and phytase-treated extrudates was not different (Table 4). OHC of the fermented extrudate was statistically significantly lower than the control, though the difference in absolute value was negligible. More than twice higher OHC values were recorded for phytase-treated and fermented powders. In this case, the difference is attributed to the powder drying method and not phytase or fermentation treatments per se, as these lyophilized powders were airy in appearance in contrast to the much denser control powder or ground extrudates. Low-density protein powders with a smaller particle size were shown to adsorb more oil than high-density powders [34]; for example, spray-dried pea protein isolates had lower OHC than lyophilized [35].

For reference, studies of pea protein isolates, which is a major constituent in the investigated blend, report WHC values in a wide range of 1.5–4.8 g H_2_O g^−1^, and OHC 1.1–5.3 g oil g^−1^, with differences attributed to pea cultivars and powder production methods [36]. It should also be noted that both OHC and WHC of extrudates were measured from ground samples. This destroys the macrostructure of meat analogs, including larger pores that could hold additional water or oil. Loosely bound water or oil in large pores can influence sensory characteristics of final products such as juiciness, moistness, and flavor release [34], thus OHC and WHC might encompass only part of the sensory perception.

Water solubility index estimates the total amount of material that can be extracted by water. Multiple factors can influence WSI, such as composition and particle size of powders, conformational state of proteins, molecular size and cross-linking. Compared to the control powder, phytase-treated powder and control extrudate were not statistically significantly different; phytase-treated extrudate had significantly lower WSI, but at a relatively small absolute difference to the control powder. This contrasts with the WHC of these samples, which was strongly influenced by extrusion. On the other hand, both fermented samples had the highest WSI due to the bacterial breakdown of macromolecules, which increased their solubility. This was also evidenced by the low viscosity of the dough seen as the low pressure at the die during extrusion. Furthermore, WSI was found to be highly negatively correlated with the die pressure (Pearson’s r −0.999).

Color change measurement helps to determine the overall impact of different treatments. As seen in Table 4, fermentation, phytase treatment, and extrusion, all had a significant effect on color. Extrusion markedly decreased lightness L*, slightly increased redness a*, but the overall effect on yellowness b* was not uniform. These results are consistent with Ilo and Berghofer [37], who concluded that L* and a* were dependent on extrusion temperature and moisture, and that lightness was the best parameter for tracking Maillard reaction. Phytase treatment darkened the powder, possibly due to the release of minerals bound to phytates. The darker tone of the phytase-treated powder was also clearly observed in extruded samples (Figure 3B), but according to the colorimetric measurement, the lowest lightness was detected in the fermented extrudate. This discrepancy arises from the fact that color measurements were performed with dried ground extrudates, whilst in Figure 3 samples are in rehydrated intact form suggesting that other parameters such as WHC or macrostructure of the extrudate influence the color of the product.

Table 5 summarizes the results of the texture profile analysis of rehydrated extrudates. The hardness relates to the maximum force required to compress the sample, chewiness is a measure of the energy required to masticate the sample, cohesiveness indicates the strength of internal bonds, springiness is how much the sample recovers after deformation, and resilience measures how fast and how strong the recovery is [38]. According to these attributes, the control and the phytase-treated meat analogs were not significantly different; in contrast, the fermented extrudate was 40% harder and 25% chewier, but at the same time slightly less cohesive. Fermented meat analog pieces were noticeably smaller in size and this lower expansion could explain their higher hardness and chewiness. Studies of expanded snacks confirm that texture properties, such as hardness, are directly related to expansion [24]. Higher dough viscosity was shown to favor higher expansion [23], whereas in this work fermented sample dough was less viscous due to macromolecular degradation and accordingly the extrudate was less expanded. Macromolecular degradation is also reflected in the lower cohesiveness of the fermented extrudate as smaller molecules formed weaker internal structures. A study of cooked chicken meat showed cohesiveness 0.37 and springiness 0.57, which are much lower than the values found in this work [39]. Interestingly, the fermented sample was the closest. Therefore further optimization of extrusion conditions is necessary for the texture of meat analogs to resemble meat more.

### 3.6. Nutritional Aspects

The nutritional quality of a protein source depends upon bioavailability, digestibility, amino acid profile, antinutrients, and other factors [40].

Phytates have a strong affinity to minerals and proteins and can hinder the absorption of nutrients. Stability of *myo*-inositol phosphate-mineral complexes has been shown to decrease with dephosphorylation of phytate, therefore improving the bioavailability of minerals [29].

The content of inositol phosphates of control powder, treated powders, and corresponding extrudates is shown in Table 6. Only IP6, IP5, and IP1 were quantified in these samples and IP4 was below the quantification limit. The most abundant form was IP6.

The lowest IP6 and IP5 content was detected in the phytase-treated and the corresponding extruded sample. IP6 was reduced by 32% and 35% from the control, respectively. Fermentation did not reduce IP6, however, IP1 concentration was 4x higher compared to the control. The same trend was also seen in small-batch spontaneous fermentation, but not in VEGE 022 (Table 2). One possible explanation for this is an enzyme with phosphatase activity that produced IP1 but could not hydrolyze IP6. Assuming that IP4, IP3, and IP2 were not detected due to some limitation of the analytical method, these inositol phosphate forms could be the substrate for the enzyme. Indeed, phosphatases with different substrate specificities and end products are well documented [41].

Extrusion degraded IP6 in the control and the fermented sample (by 18% and 8%, respectively), whereas IP5 content was significantly higher only in the fermented sample. This supports the results of Alonso et al. (2001) and Pontoppidan et al. (2007) [14,42]. In these experiments high temperature and pressure during extrusion cooking degraded IP6 to lower phosphate forms and this reduction depended on the extruded material.

The content of essential amino acids defines the nutritional quality of a protein. The Food and Agriculture Organization of the United Nations (FAO) has established a standard for protein quality evaluation that sets a minimum amount of essential amino acids per gram of total protein for different age groups [43]. Plant proteins are regarded as nutritionally incomplete: cereals are typically deficient in lysine and legumes are low in methionine and cysteine [40]. This is seen in Table 7 that shows the amino acid composition of pea and oat proteins and their blend. According to the FAO recommendation for infants, both protein concentrates are nutritionally incomplete. Yet, the content of essential amino acids of pea isolate is above the recommended minimum for adults, while oat concentrate is deficient only in lysine. Therefore, powder blending improved nutritional quality, and the control blend fulfilled FAO recommendations for adults.

The free amino acid content is summarized by sparse principal component analysis in Figure 4. The largest difference is seen between the fermented and other samples, as captured by the first principal component. Fermentation is known to increase free amino acids due to proteolytic activity [44,45]. In this work, the total free amino acid content of the fermented sample was twice higher compared to the control (1.77 ± 0.12 and 0.85 ± 0.08 mg g^−1^ dwb). Fermentation increased content of free Ala, Cys, His, Ile, Leu, Lys, Phe, Pro, Ser, Tyr, and Val, but decreased Arg, Asn, and Asp. The second principal component represents the effect of extrusion, which was much smaller than fermentation. Extrusion cooking mostly had reduced the availability of free Gln, Met, and Trp; the effect on Gly was not statistically significant. Small soluble molecules such as free amino acids contribute to WSI (Table 4). This is corroborated by a high Pearson’s correlation coefficient of 0.929.

### 3.7. Sensory Assessment

The sensory assessment of extrudates is presented as a radar diagram in Figure 5. The difference between the control and the phytase-treated sample was small, as evidenced by the overlap of the lines. The phytase-treated extrudate tended to be significantly harder, more intense in color, and had stronger off-odor in comparison to the control. Interestingly, the assessors commented on this sample as having sweet, bready, roasty, and ashy odors. The fermented sample was more clearly distinguished from the other two extrudates. The following attributes were statistically significantly higher in the fermented sample compared to the control: overall taste intensity, sour odor and taste, cereal and legume taste, bitter taste, off-odor and off-taste intensity, and aftertaste intensity. On the other hand, springiness, chewiness, cohesiveness, and moistness were graded lower. Some assessors described fermented samples as being dry and easily disintegrating and noted medicinal, fermented, soapy, and citrus off-flavors.

The low sensorial moistness of the fermented extrudate was reflected in its low water holding capacity (Table 4). Yet, other sensory texture attributes had a very low correlation with the texture profile analysis results underlining the difficulty of evaluating this type of material.

Fermentation developed flavor compounds that enhanced the overall taste profile, but some attributes, such as cereal or bitter taste, can be considered negative or unwanted in a meat analog product. In contrast to these results, Rani et al. (2018) [44] demonstrated that puffed snacks prepared from fermented rice-black gram mixed flour had superior acceptability as fermentation improved sensory attributes of texture, mouthfeel, taste, and aftertaste. Thus it can be understood, that fermentation has to be optimized to avoid the formation of unwanted compounds.

### 3.8. Volatile Compounds

Volatiles are represented by a partial least squares discriminant analysis in Figure 6. As evident from the plot, the control powder used in this trial correlated with most of the identified volatile compounds. These compounds can all be traced back to the raw materials—pea and oat—and can be considered as remnants of protein powder production [46,47]. The amount of these compounds, however, noticeably decreased as a result of the extrusion process which can be primarily attributed to the use of heat (up to 150 °C). This compares to some of the earlier studies where the application of heat during extrusion has been tied to compositional losses [48,49].

The extruded samples were different from the control powder due to an increase in various pyrazines, thiophenes (thiophene, 2-methyl-; thiophene, 2-pentyl-; thiophene, 2-hexyl-), furans (furfural, 2-furanmethanol), and 1-pentanethiol. These compounds could all be viewed as products of the Maillard reaction. The latter could also have formed as a result of interaction between the by-products of Maillard reaction and lipid oxidation [50].

In the Maillard reaction, reductive sugars react with amino acids to produce brown pigments (melanoidins) and characteristic volatile compounds [51]. Volatile compounds produced as a result include but not limited to pyrazines, pyridines, pyrroles, furans, furanones, pyranones, oxazoles, thiophenes [52]. The Maillard reaction compounds associated with extrusion were previously found to depend on the initial amino acid composition of the extrudate. For example, increased production of sulfur compounds was noted in extrudates supplemented with cysteine [53]. The formation of thiophenes and furans could also be tied to specific amino acids contained in the samples: the formation of thiophenes could be attributed to methionine and its inclusion in the Maillard reaction pathways; the formation of furans—to alanine [54,55]. The increase in pyrazines was also previously connected to individual amino acid supplementation [56]. Hence the fermented extrudate correlated with most of the identified pyrazines due to the increase in free amino acid content. Additionally, higher production of pyrazines in the fermented sample could be attributed to its lower pH, corroborating other studies [57].

The phytase-treated sample can be considered similar to the extruded control which means that phytase action did not add to the release of volatile compounds or their precursors. The fermented extruded sample, on the other hand, additionally contained elevated amounts of carboxylic acids (butanoic acid; butanoic acid, 2-methyl-; butanoic acid, 3-methyl-; pentanoic acid; hexanoic acid; heptanoic acid; nonanoic acid), alcohols (1-hexanol, benzyl alcohol), acetoin, and 2,4-decadienal. These can be considered as products of metabolic activity of the chosen starter culture (VEGE 053) that consists primarily of lactic acid bacteria (*Streptococcus*, *Lactobacillus*, *Lactococcus*) and bifidobacteria [58,59]. The higher amount of carboxylic acids in this sample correlated with the sourness in both perceived odor and taste. This could have also contributed to the “fermented” off-flavor perceived by the panel. The off-flavor described as ”medicinal”, “soapy”, and “citrus-like” could be attributed to the presence of 2,4-decadienal in this sample (based on www.thegoodscentscompany.com database description for this compound).

## 4. Conclusions

Extrusion was successfully applied to produce fibrous meat analogs from the pea-oat protein blend. This blend fulfills minimum recommended amounts of essential amino acids for adults according to FAO; however, it contains 1.5% of phytic acid, a known antinutrient. The addition of a phytase even in non-optimal conditions has reduced phytates by 30%. Moreover, it was demonstrated that extrusion also partially degraded phytates in a matrix-dependent manner, thus improving the nutritional quality of the material. Fermentation had a strong effect on the extrusion process and physicochemical, textural, and sensorial properties of extrudates. Fermented material required much stronger treatment intensity to form fibrous structure and sensory assessment of extrudates showed an intensification of flavors, though some of the taste attributes could be considered undesired in a meat analog. Yet, fermentation did not reduce phytic acid as the starters used in the experiment had no phytase activity. This suggests that further optimization of the fermentation process and selection of starters is necessary to obtain products with reduced phytic acid content and improved sensory characteristics.

## Figures and Tables

**Figure 1 foods-09-01059-f001:**
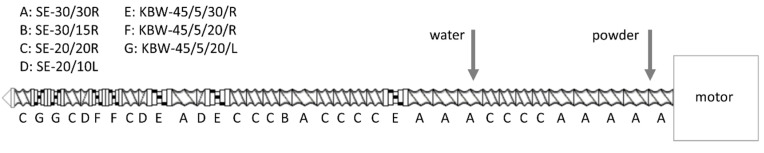
Screw configuration. The screw diameter is 20 mm. Single letters correspond to the manufacturer’s screw part codes as presented in the top left corner, where SE—conveying element, KBW—kneading block, R—forward element, L—reverse element, and numbers encode angle (KBW only), step and element lengths in mm.

**Figure 2 foods-09-01059-f002:**
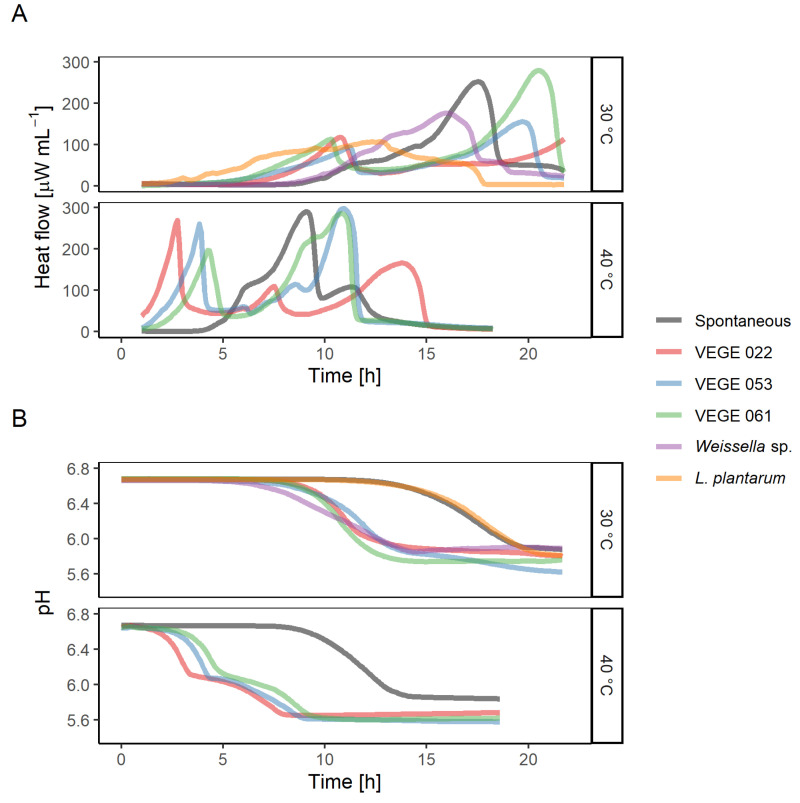
Small-scale fermentation by the starter cultures at 30 and 40 °C. The spontaneous sample was not inoculated. (**A**) Typical microcalorimetry heat flow curves. (**B**) pH change.

**Figure 3 foods-09-01059-f003:**
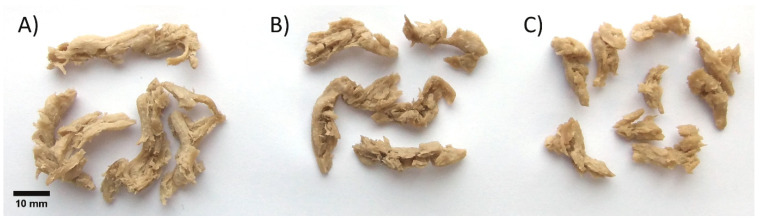
Rehydrated extruded materials from (**A**) control pea-oat powder, (**B**) phytase-treated powder, (**C**) fermented powder.

**Figure 4 foods-09-01059-f004:**
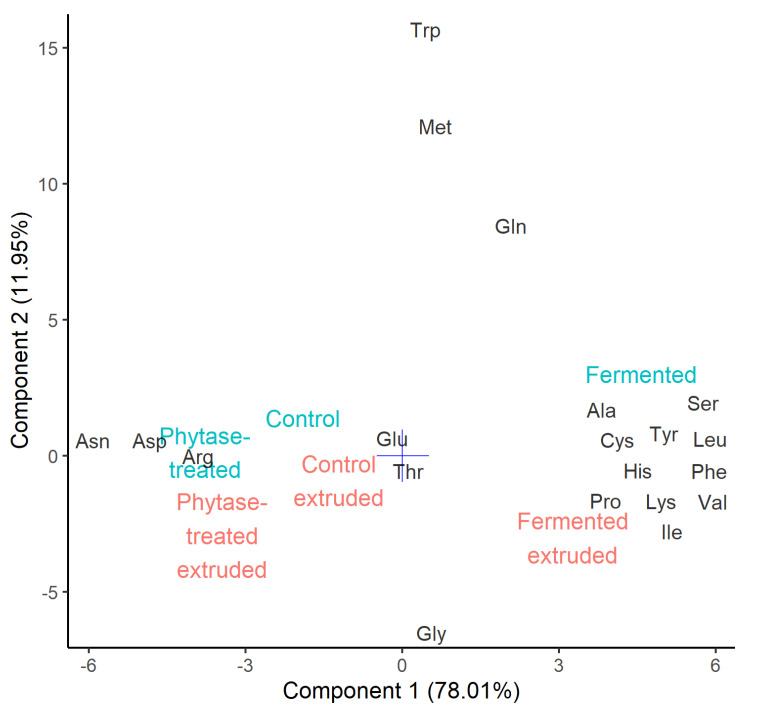
Sparse principal component analysis of free amino acid content of powders (blue) and extrudates (red). The number in parenthesis indicates variance captured by the principal component.

**Figure 5 foods-09-01059-f005:**
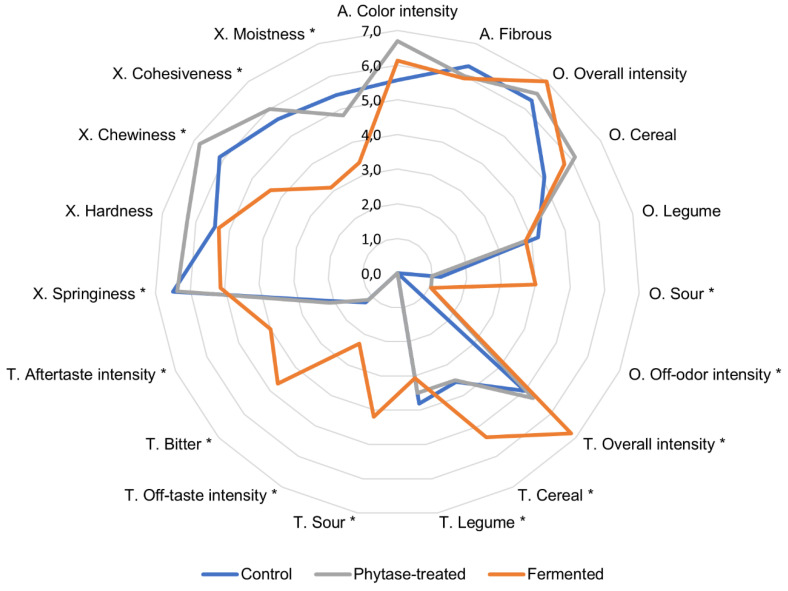
Sensory analysis of extrudates. Here, A means appearance, O—odor, T—taste, and X—texture. An asterisk indicates a statistically significant difference between the control and the fermented sample.

**Figure 6 foods-09-01059-f006:**
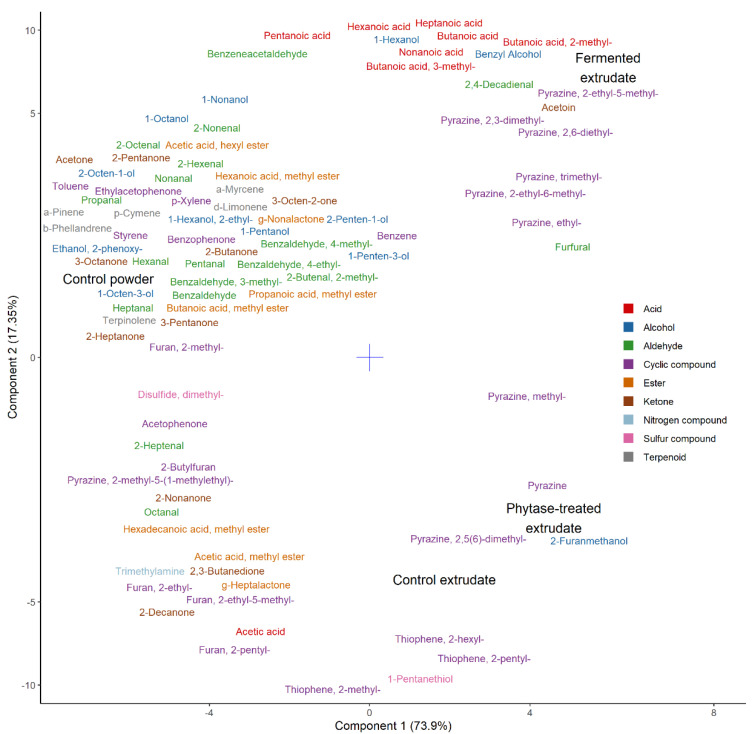
Partial least squares discriminant analysis of gas chromatography data. The number in parenthesis indicates variance captured by the principal component.

**Table 1 foods-09-01059-t001:** Total phytates content [% dry weight basis (dwb)] measured with the Megazyme kit in powders fermented by different starter cultures. Standard deviations are shown (*n* = 3). Values sharing a letter are not statistically different.

Starter	Total Phytates [% dwb]
Control	1.42 ± 0.03 ^c^
Spontaneous, 40 °C	1.92 ± 0.01 ^a^
VEGE 022, 40 °C	1.51 ± 0.01 ^b^
VEGE 053, 40 °C	1.50 ± 0.01 ^b^
VEGE 061, 40 °C	1.52 ± 0.01 ^b^
*Weissella* sp., 30 °C	1.46 ± 0.02 ^bc^
*L. plantarum*, 30 °C	1.95 ± 0.05 ^a^

**Table 2 foods-09-01059-t002:** Inositol phosphates content [% dwb] measured with the LC-MS method in powders fermented by different starter cultures. Standard deviations are shown (*n* = 3). Values sharing a letter within a column are not statistically different.

Starter	IP6	IP5	IP1
Control	1.29 ± 0.08 ^a^	0.12 ± 0.01 ^a^	0.04 ± 0.00 ^b^
Spontaneous, 40 °C	1.22 ± 0.03 ^a^	0.11 ± 0.02 ^a^	0.21 ± 0.01 ^a^
VEGE 022, 40 °C	1.20 ± 0.02 ^a^	0.11 ± 0.01 ^a^	0.04 ± 0.00 ^b^

**Table 3 foods-09-01059-t003:** Actual measured extrusion parameters. Standard deviations are shown (*n* = 180). Values sharing a letter within a row are not statistically different. SME: specific mechanical energy.

	Control	Phytase-Treated	Fermented
Temperature at the die [°C]	149.0 ± 0.1 ^c^	149.2 ± 0.1 ^b^	148.0 ± 0.9 ^a^
Pressure at the die [bar]	17.0 ± 0.8 ^a^	15.8 ± 1.0 ^b^	10.5 ± 1.0 ^c^
Dough moisture [%]	31.2	31.6	27.8
SME [Wh kg^−1^]	162	157	174

**Table 4 foods-09-01059-t004:** Physicochemical properties of powders and extrudates. Values sharing a letter within a row are not statistically different. WHC: water holding capacity; OHC: oil holding capacity; WSI: water solubility index; L*, a*, b*: CIELAB color space values.

	Powders			Extrudates		
	Control	Phytase-Treated	Fermented	Control	Phytase-Treated	Fermented
WHC [g H_2_O g^−1^]	2.1 ^a^	2.0 ^a^	1.4 ^b^	1.5 ^b^	1.3 ^b^	1.2 ^b^
OHC [g oil g^−1^]	0.8 ^b^	1.9 ^a^	1.8 ^a^	0.8 ^b^	0.8 ^b^	0.7^c^
WSI [%]	10.9 ^c^	11.5 ^c^	14.7 ^a^	11.6 ^c^	9.9 ^d^	13.7 ^b^
L*	84.3 ^a^	77.2 ^c^	81.6 ^b^	73.3 ^d^	72.1 ^e^	70.4 ^f^
a*	2.8 ^c^	1.9 ^d^	3.3 ^b^	3.3 ^b^	4.6 ^a^	4.7 ^a^
b*	21.4 ^c^	19.0 ^e^	22.5 ^b^	20.3 ^d^	21.1 ^c^	23.4 ^a^

**Table 5 foods-09-01059-t005:** Texture profile analysis of the extrudates. Values sharing a letter within a row are not statistically different. Standard deviations are shown (*n* = 12).

	Control	Phytase-Treated	Fermented
Hardness [N]	29 ± 6 ^b^	34 ± 5 ^b^	49 ± 6 ^a^
Chewiness [N]	17 ± 4 ^b^	18 ± 3 ^b^	23 ± 3 ^a^
Cohesiveness	0.64 ± 0.01 ^a^	0.63 ± 0.03 ^a^	0.57 ± 0.02 ^b^
Springiness	0.91 ± 0.06 ^a^	0.84 ± 0.10 ^a^	0.83 ± 0.09 ^a^
Resilience	0.42 ± 0.02 ^a^	0.43 ± 0.03 ^a^	0.40 ± 0.03 ^a^

**Table 6 foods-09-01059-t006:** Inositol phosphates content [% dwb] measured with the LC-MS method in treated powders and extrudates. Standard deviations are shown (*n* = 3). Values sharing a letter within a column are not statistically different.

Sample	IP6	IP5	IP1
Control	1.31 ± 0.02 ^a^	0.11 ± 0.01 ^b^	0.05 ± 0.01 ^c^
Phytase-treated	0.89 ± 0.04 ^d^	0.05 ± 0.01 ^c^	0.09 ± 0.02 ^b^
Fermented	1.30 ± 0.02 ^a^	0.08 ± 0.01 ^c^	0.20 ± 0.01 ^a^
Control extruded	1.07 ± 0.05 ^c^	0.14 ± 0.01 ^ab^	0.04 ± 0.01 ^c^
Phytase-treated extruded	0.85 ± 0.02 ^d^	0.06 ± 0.01 ^c^	0.07 ± 0.01 ^bc^
Fermented extruded	1.19 ± 0.03 ^b^	0.16 ± 0.01 ^a^	0.18 ± 0.03 ^a^

**Table 7 foods-09-01059-t007:** Total amino acid content of pea protein isolate, oat protein concentrate, control blend [mg g^−1^ protein], and the recommended minimum amount by FAO [43].

Amino Acid	Pea Protein	Oat Protein	Control	FAO Recommended for Adults	FAO Recommended for Infants
Ala	40	42	41		
Arg	99	68	89		
Asp	144	75	124		
Cys	15	25	18		
Glu	190	269	214		
Gly	41	39	40		
His	23	19	22	16	21
Ile	43	38	42	30	55
Leu	78	78	78	61	96
Lys	71	35	60	48	69
Met	14	17	15		
Phe	51	52	51		
Pro	42	53	45		
Ser	52	40	48		
Thr	36	30	34	25	44
Trp	10	13	11	7	17
Tyr	70	40	61		
Val	48	51	49	40	55
Cys + Met	29	42	33	23	33
Phe + Tyr	121	92	112	41	94

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
