# Peer review of "Impact of Fermentation and Phytase Treatment of Pea-Oat Protein Blend on Physicochemical, Sensory, and Nutritional Properties of Extruded Meat Analogs"

_foods, 2020, doi:10.3390/foods9081059_

Round 1

Reviewer 1 Report

The paper titled: ,,Impact of fermentation and phytase treatment of pea- oat protein blend on physicochemical, sensory, and nutritional properties of extruded meat analogs” sent by Aleksei Kaleda, Karel Talvistu, Martti Tamm, Maret Viirma, Julia Rosend, Kristel Tanilas, Marie Kriisa, Natalja Part  and Mari-Liis Tammik investigates the impact of fermentation process and phytase treatment of pea-oat protein blend (extruded) on many properties of final products (extruded - phytase treatment and fermented). The article is very interesting, but I have a few suggestions.

Materials: what was the mass ratio of pea and oat proteins in the mixture?

Discussion about oil holding capacity, water holding capacity, and water solubility index is too short, please add some more information about comparison between powders and extrudates and about structure of that products.

Please explain, why in Table 5 the authors describe only extrudates, not powders like in Table 4? The discussion about texture profile is very poor, please expand it.

In table 7 the authors presented amino acids profile for pea, oat proteins and control sample. Please explain, why the results of fermented and phytase treatment samples are not presented in the table? Figure 4 is a bit unreadable. In my opinion it would be more appropriate to summarize all the results in one table.

Line 350 – 352, page 9: ,,The control and the phytase-treated meat analogs were not significantly different; in contrast, the  fermented extrudate was harder and chewier, but at the same time slightly less cohesive. This can be observed in Figure 3C, as the fermented pieces are smaller and denser.” Should these parameters be distinguished based on the photos? The wording of this sentence is unfortunate.

The discussion about volatile compounds is insufficient and Figure 6 is illegible.

The article will be suitable for publication after major amendments.

Reviewer 2 Report

I think this manuscript is interesting and valuable research.

Since there has been little research regarding the properties of fermented (or phytase treated) plant proteins as meat analog, I feel this study is original and has a good novelty.

The selected treatments would modify physicochemical and sensorial properties of meat analogue compring to normal processed one, and the authors explained the background of it in Introduction.

Throughout the manuscript, this study described both nutritional and processing aspects of meat analogue produced by fermented (or phytase treated) pea-oat proteins.

The experiment is properly designed and results are supported by logical discussion.

Just my concern is that there are citations of too many references.

Even in Introduction, total 38 references were cited, and I suggest to authors to delete some references which are not imporatant or not directly-related with this study before final acceptance.

Minorly, there were many light fonts in Figure 6, so it would be better to change the fonts with more clear color.

Very interesting investigation and this study has valuable from the academic and industrial application point of views.

Current manuscript is well written with logical explanation, and I feel this manuscript cab be published as it is.

Just minor change can improve readability of this study, but it doesn't affect my recommendation of this manuscript.

  • If possible, please improve resolution of figure 6 (please use clear pont color).
  • Total 71 references look like too many comparing the length of the manuscript. Please delete unnessessary references.
